# Tree Ensembles for Contextual Bandits

**Hannes Nilsson**[*]                                                      *hannesni@chalmers.se*
*Chalmers University of Technology and University of Gothenburg*

**Rikard Johansson**[*]                                                      *rikjo@chalmers.se*
*Chalmers University of Technology and University of Gothenburg*

**Niklas Åkerblom**                                                *niklas.akerblom@volvocars.com*
*Volvo Car Corporation*
*Chalmers University of Technology and University of Gothenburg*

**Morteza Haghir Chehreghani**                              *morteza.chehreghani@chalmers.se*
*Chalmers University of Technology and University of Gothenburg*

**Reviewed on OpenReview:** *https://openreview.net/forum?id=59DCkSGw8S*

## Abstract

We propose a new framework for contextual multi-armed bandits based on tree ensembles. Our framework adapts two widely used bandit methods, Upper Confidence Bound and Thompson Sampling, for both standard and combinatorial settings. As part of this framework, we propose a novel method of estimating the uncertainty in tree ensemble predictions. We further demonstrate the effectiveness of our framework via several experimental studies, employing XGBoost and random forests, two popular tree ensemble methods. Compared to state-of-the-art methods based on decision trees and neural networks, our methods exhibit superior performance in terms of both regret minimization and computational runtime, when applied to benchmark datasets and the real-world application of navigation over road networks.

## 1 Introduction

Stochastic *multi-armed bandits* (MABs) (see Slivkins, 2019) provide a principled framework for making optimal sequences of decisions under uncertainty. As discussed by Zhu & Van Roy (2023), MABs constitute a vital component of modern recommendation systems for efficient exploration, among many other applications. An important variant known as the *contextual multi-armed bandit* incorporates additional contextual / side information into the decision-making process, allowing for more personalized or adaptive action selection. Following the recent success of (deep) neural networks in solving various machine learning tasks, several methods building on such models have been suggested for finding functional relationships between the context and outcomes of actions to aid the decision-making process (Zhou et al., 2020; Zhang et al., 2021; Zhu & Van Roy, 2023; Osband et al., 2021; Hoseini et al., 2022). Even though sophisticated, these methods can be impractical, time-consuming, and computationally expensive.

In light of these challenges, we propose bandit algorithms that utilize *tree ensemble* (TE) methods (Hastie et al., 2009), like *gradient-boosted decision trees* (GBDT) and *random forests*, to comprehend the contextual features, combined with the most popular bandit methods *Upper Confidence Bound* (UCB) and *Thompson Sampling* (TS) as exploration strategies. Compared with other tree-based and neural network methods, our methods, called TEUCB and TETS, yield superior results on UCI benchmark datasets. Additionally, our methods benefit from more effective learning with less computational overhead on most problem instances, compared to existing methods that use similarly expressive machine learning models. We furthermore

---

[*]These authors contributed equally.

extend the framework to the combinatorial contextual setting, a type of bandit that deals with complex combinatorial sets of arms, and investigate it on the important real-world application of efficient navigation in road networks. In this work, we focus on the practical applicability of the proposed framework in different settings and do not provide a theoretical analysis of the regret.

## 1.1 Related Work

The concept of applying decision trees to contextual bandit problems has been studied to a limited extent. Elmachtoub et al. (2017) suggest using a single decision tree for each arm available to the agent. Their method is referred to as *TreeBootstrap*, as the concept of bootstrapping is employed in order to resemble the Thompson Sampling process. One issue with their method is its limited ability to achieve the accuracy and robustness of tree ensemble methods, particularly when estimating complex reward functions. The possibility of applying random forests in their framework is discussed, but no concrete methods or experimental results are presented. Furthermore, storing and training one tree model per arm does not scale well, especially not with large action spaces (e.g., in the combinatorial setting), and could potentially lead to excessive exploration for dynamic arm sets, as it cannot attend to what it has learned from the contexts of other arms. In contrast, Féraud et al. (2016) employ a tree ensemble but only use decision trees of unit depth, also known as decision stumps. While tree ensembles with a large number of decision stumps tend to perform well regarding accuracy, lower variance, and increased robustness, restricting the depth in such an extreme way may not be adequate when addressing complex tasks (Hastie et al., 2009). The experimental studies by Hastie et al. (2009) indicate that tree depths of 4 to 8 work well for boosting. Additionally, Féraud et al. (2016) only investigate the bandit algorithm known as *successive elimination*, which usually leads to less efficient exploration compared to UCB and TS, and thus is used less commonly. Moreover, the method, called *Bandit Forest*, requires binary encoded contextual features. Comparing these two tree-based bandit methods, Elmachtoub et al. (2017) experimentally show that TreeBootstrap tends to outperform Bandit Forest in practice. It should be noted that neither Elmachtoub et al. (2017) nor Féraud et al. (2016) consider combinatorial contextual bandits.

Apart from the two methods mentioned above, the problem has primarily been addressed using other machine learning models. When the reward function is linear w.r.t. the context, *LinUCB* (Li et al., 2010; Abbasi-Yadkori et al., 2011) and *Linear Thompson Sampling* (LinTS) (Agrawal & Goyal, 2013) have demonstrated good performance. In more complicated cases, the linear assumption no longer holds, and thus more expressive models are needed.

In recent years, neural contextual bandits have attracted a lot of attention. As the name suggests, these methods use (deep) neural networks to model the expected rewards for some observed contextual features. Out of several proposed neural contextual bandit algorithms, *NeuralUCB* (Zhou et al., 2020) and *NeuralTS* (Zhang et al., 2021), in particular, have been shown to perform well, while also providing theoretical regret bounds. On the negative side, due to how these methods estimate the uncertainty in predictions, they rely on inverting a matrix of size equal to the number of network parameters, which is usually computationally expensive and time-consuming, especially when the prediction tasks require large neural networks.

Zhu & Van Roy (2023) provide a thorough review of different neural bandit methods and suggest a method based on epistemic neural networks (Osband et al., 2021) for more efficient uncertainty modeling, showing its advantages in terms of regret minimization and computational efficiency. Despite their promising results, we argue that neural networks may not necessarily be the most appropriate models for contextual bandits. For instance, even with epistemic neural networks, training is still resource-intensive. On the other hand, there is a large body of work that shows the effectiveness of tree ensemble methods for regression and classification tasks in the standard supervised learning setting (Borisov et al., 2022; Gorishniy et al., 2021; Grinsztajn et al., 2022), and therefore we believe that their extension to contextual bandits can have great potential.

One less directly related method is *Ensemble Sampling* (Lu & Van Roy, 2017), where the ensembles are solely used for uncertainty estimation and not for modeling the reward functions in bandits. Uncertainty estimates in Ensemble Sampling and similar methods focus on uncertainties in the parameters of the models used to approximate the underlying processes being sampled. In contrast, we sample directly from estimated

distributions of the expected rewards given contexts. Our approach aligns closer with sampling-based bandit methods like LinTS and NeuralTS.

## 2 Background

In this section, we describe the multi-armed bandit problem and its extensions relevant to our work.

### 2.1 Multi-Armed Bandit Problem

The multi-armed bandit (MAB) problem is a sequential decision-making problem under uncertainty, where the decision maker, i.e., the agent, interactively learns from the environment over a horizon of $T$ time steps (interactions). At each time step $t \leq T$, the agent is presented with a set of $K$ actions $\mathcal{A}$, commonly referred to as arms. Each action $a \in \mathcal{A}$ has an (initially) unknown reward distribution with expected value $\mu_a$. The objective is to maximize the cumulative reward over the time horizon $T$, or more commonly (and equivalently) to minimize the cumulative regret, which is defined as

$$R(T) \triangleq \sum_{t=1}^{T} (\mu_{a^*} - \mu_{a_t}), \tag{1}$$

where $\mu_{a^*}$ is the expected reward of the optimal arm $a^*$, and $a_t$ is the arm selected at time step $t$. It is important to note that the agent only receives feedback from the selected arm/action. More specifically, when selecting $a_t$ the agent receives a reward $r_{t,a_t}$ which is sampled from the underlying reward distribution. As previously mentioned, the agent's objective is to minimize the cumulative regret, not to learn the complete reward function for each arm. Thus, the agent has to balance delicately between exploration and exploitation.

### 2.2 Contextual Bandit Problem

The contextual multi-armed bandit problem is an extension of the classical MAB problem described in Section 2.1. The agent is, at each time step $t$, presented with a context vector $\mathbf{x}_{t,a} \in \mathbb{R}^d$ for each action $a \in \mathcal{A}$. For example, a recommendation system could encode a combination of user-related and action-specific features into the context vector $\mathbf{x}_{t,a}$. Then, the expected reward of action $a$ at time $t$ is given by an (unknown) function of the context $q : \mathbb{R}^d \to \mathbb{R}$, such that $\mathbb{E}[r_{t,a}] = q(\mathbf{x}_{t,a})$. Learning to generalize the relationship between the contextual features and the expected reward is crucial for effective learning and minimizing cumulative regret.

### 2.3 Combinatorial Bandits

Combinatorial multi-armed bandits (CMAB) (Cesa-Bianchi & Lugosi, 2012) deal with problems where at each time step $t$, a subset $\mathcal{S}_t$ (called a *super arm*) of the set of all *base arms* $\mathcal{A}$ is selected, instead of an individual arm. The reward of a super arm $\mathcal{S}$ depends on its constituent base arms. When the reward for a super arm is given as a function (e.g., sum) of feedback from its individual base arms (observable if and only if the super arm is selected), it is referred to as *semi-bandit feedback*. As previously, the evaluation metric is the cumulative regret, and for the combinatorial semi-bandit setting (with the sum of base arm feedback as super arm reward) it can be defined as

$$R(T) \triangleq \sum_{t=1}^{T} \left( \sum_{i \in \mathcal{S}_t^*} \mu_i - \sum_{j \in \mathcal{S}_t} \mu_j \right), \tag{2}$$

where $\mathcal{S}_t^*$ denotes the optimal super arm at time $t$.

## 3 Proposed Algorithms

Machine learning models based on decision trees have consistently demonstrated solid performance across various supervised learning tasks (Borisov et al., 2022; Gorishniy et al., 2021; Grinsztajn et al., 2022). An

up-to-date overview of tree ensemble methods is provided by Blockeel et al. (2023), where several advantages are discussed over other techniques. For instance, they are known to learn fast from small sets of examples, while simultaneously possessing the capability of handling large data sets, and are computationally efficient to train.

Despite the potential benefits, these types of models have not been studied much for contextual bandits. To the best of our knowledge, there is no previously known work that in a principled way combines tree ensemble models with UCB, which is one of the most effective strategies known for handling the exploration-exploitation dilemma. Further, works that combine tree ensemble methods with Thompson Sampling are limited. We address this research gap and introduce a novel approach for the contextual MAB problem that combines any tree ensemble method with a natural adaption of these two prominent exploration schemes.

Within our framework, we empirically investigate the XGBoost and random forest algorithms and demonstrate promising performance on several benchmark tasks, as well as a combinatorial contextual MAB problem for efficient navigation over a stochastic road network in Section 4. However, we emphasize that our bandit algorithms are sufficiently general to be employed with any decision tree ensemble.

### 3.1  Tree-Based Weak Learners

The underlying concept of decision tree ensembles is to combine several *weak learners*. Each standalone tree is sufficiently expressive to learn simple relationships and patterns in the data, yet simple enough to prevent overfitting to noise. By combining the relatively poor predictions of a large number of weak learners, the errors they all make individually tend to cancel out, while the true underlying signal is enhanced.

In our notation, a tree ensemble regressor $f$ is a collection of $N$ decision trees $\{f_n\}$, where the trees are collectively fitted to a set of training samples $\{(\mathbf{x}, r)\}$. Each sample consists of a context vector $\mathbf{x}$ and a target value $r$. The fitting procedure can differ between different types of tree ensemble methods, which are often divided into two categories based on the approach used, i.e., *bagging* or *boosting*. In both variants, the prediction of each tree is determined by assigning the training samples to distinct leaves, where all samples in the same leaf resemble each other in some way, based on the features attended to in the splitting criteria associated with the ancestor nodes of the leaf. Hence, when fitted to the data, each tree $f_n$ receives an input vector $\mathbf{x}$ which it assigns to one of its leaves.

Every leaf is associated with three values that depend on the samples from the training data assigned to that particular leaf. We denote the output of the $n$th tree by $o_n$, which is this tree's contribution to the total ensemble prediction and depends on which leaf the input vector $\mathbf{x}$ is assigned to. The number of training samples assigned to the same leaf as $\mathbf{x}$ in tree $n$ is denoted by $c_n$. Finally, the sample variance in the output proposed by the individual training samples assigned to the leaf is denoted $s_n^2$. This value represents the uncertainty in the output $o_n$ associated with the leaf and depends not only on the training samples — but also on the particular tree ensemble method used. We elaborate on the calculations of $s_n^2$ for two types of tree ensembles in Sections 3.6 and 3.7. Note that, for the sample variance to exist, we must have at least two samples. This is ensured by requiring the tree to be built such that no leaf has fewer than two training samples assigned to it.

The tree ensemble constructs its total target value prediction $p$ by summing up all of the $N$ individual trees' outputs $o_n$:

$$p(\mathbf{x}) = \sum_{i=1}^{N} o_i(\mathbf{x}). \tag{3}$$

The outputs $o_n$, in turn, are averages of the $c_n$ outputs suggested by each training sample assigned to the same leaf. These suggested outputs are based on the target values of the training samples, with the calculation method depending on the specific type of tree ensemble used. We always consider the full tree ensemble target value predictions as a sum of the individual tree predictions $o_n$ to unify the notation of all kinds of tree ensembles. Hence, for averaging models, such as random forests, we assume that the leaf values $o_n$ are already weighted by $1/N$.

### 3.2 Uncertainty Modeling

For bandit problems, keeping track of the uncertainty in the outcomes of different actions is crucial for guiding the decision-making process. In order to form estimates of the uncertainty in the final prediction of the tree ensembles we employ, we make a few assumptions.

Firstly, we assume that the output $o_n$ of the $n$'th decision tree, given a context $\mathbf{x}$, is an arithmetic average of $c_n$ independent and identically distributed random variables with finite mean $\mu_n$ and variance $\sigma_n^2$. By this assumption, $o_n$ is itself a random variable with mean $\mu_n$ and variance $\frac{\sigma_n^2}{c_n}$. Also, we can approximate $\mu_n$ and $\sigma_n^2$ by the sample mean and variance respectively, based on the training samples assigned to the leaf. Moreover, the central limit theorem (CLT) (see e.g., Dodge, 2008) ensures that, as $c_n \to \infty$, the distribution of $o_n$ tends to a Gaussian distribution. See Appendix A.2 for more details about the independence assumption.

Secondly, if we also assume the output of each one of the $N$ trees in the ensemble to be independent of every other tree, we have that the total tree ensemble's target value prediction, which is a sum of $N$ random variables, is normally distributed with mean $\mu = \sum_{i=1}^{N} \mu_i$ and variance $\sigma^2 = \sum_{i=1}^{N} \sigma_i^2$.

We acknowledge that these assumptions may not always be true in practice, but they act here as motivation for the design of our proposed algorithms. Specifically, they entail a straightforward way of accumulating the variances of the individual tree contributions to obtain an uncertainty estimate in the total reward prediction, allowing us to construct efficient exploration strategies. In the following two subsections, we present our approach to doing so with UCB and TS methods, respectively.

### 3.3 Tree Ensemble Upper Confidence Bound

UCB methods act under the principle of *optimism in the face of uncertainty*, and have been established as some of the most prominent approaches to handling exploration in bandit problems. A classic example is the UCB1 algorithm (Auer et al., 2002) which builds confidence bounds around the expected reward from each arm depending on the fraction of times it has been played, and for which there are proven upper bounds on the expected regret. One disadvantage of the method, however, is that it does not take the variance of the observed rewards into account, which may lead to a sub-optimal exploration strategy in practice. In light of this, Auer et al. (2002) further proposed the UCB1-Tuned and UCB1-Normal algorithms, which extend UCB1 by including variance estimates, and demonstrate better performance experimentally. The main difference between the two extended versions is that UCB1-Tuned assumes Bernoulli-distributed rewards, while UCB1-Normal is constructed for Gaussian rewards. At each time step $t$, UCB1-Normal selects the arm $a$ with the maximal upper confidence bound $U_{t,a}$, calculated as

$$U_{t,a} \leftarrow \tilde{\mu}_{t,a} + \sqrt{16\tilde{\sigma}_{t,a}^2 \frac{\ln(t-1)}{m_{t,a}}}, \tag{4}$$

where $m_{t,a}$ is the number of times arm $a$ has been played so far, and $\tilde{\mu}_{t,a}$ and $\tilde{\sigma}_{t,a}^2$ are the sample mean and variance of the corresponding observed rewards, respectively. For the sample variance to be defined for all arms, they must have been played at least twice first.

By the assumptions we have made on the tree ensembles, the predictions of the expected rewards will be approximately Gaussian. Therefore, we propose an algorithm called *Tree Ensemble Upper Confidence Bound* (TEUCB) in Algorithm 1, which draws inspiration from UCB1-Normal, and is suitable for both Gaussian and Bernoulli bandits. As seen in lines 27 and 31 of Algorithm 1, the selection rule of TEUCB closely resembles that of UCB1-Normal, but is constructed specifically for contextual MABs where the arms available in each time step are characterized by their context vectors. Therefore, TEUCB considers each individual contribution from all samples in a leaf as a sample of that leaf's output distribution, which is demonstrated in line 14. The total prediction for a context $\mathbf{x}_{t,a}$ is subsequently computed as the sum of sampled contributions from each leaf that $\mathbf{x}_{t,a}$ is assigned to in the ensemble (lines 16, 17, 18).

Beyond yielding better performance in experiments, there are additional benefits associated with considering the sample variances of rewards in the TEUCB method, as discussed regarding UCB1-Normal. Since each tree in the ensemble may be given a different weight, certain trees can contribute more than others to the

---

**Algorithm 1** Tree Ensemble Upper Confidence Bound / Tree Ensemble Thompson Sampling

---

1: **Input:** Number of rounds $T$, number of initial random selection rounds $T_I$, number of trees in ensemble $N$, exploration factor $\nu$, tree ensemble regressor $f$, bandit method (UCB or TS).
2: **for** $t = 1$ **to** $T_I$ **do**
3:     Randomly select and play an arm $a_t$
4:     Observe context $\mathbf{x}_{t,a_t}$ and reward $r_{t,a_t}$
5: **end for**
6: **for** $t = T_I + 1$ **to** $T$ **do**
7:     Fit tree ensemble $f$ to previously observed context-reward pairs $\{(\mathbf{x}_{i,a_i}, r_{i,a_i})\}_{i=1}^{t-1}$ and set leaf values
8:     Observe current contexts $\{\mathbf{x}_{t,a}\}_{a=1}^K$
9:     **for** $a = 1$ **to** $K$ **do**
10:         Initialize arm parameters:
            $\tilde{\mu}_{t,a} \leftarrow 0, \quad \tilde{\sigma}_{t,a}^2 \leftarrow 0, \quad c_{t,a} \leftarrow 0$
11:         **for** n = 1 **to** $N$ **do**
12:             Assign leaf values:
13:             $(o_{t,a,n}, s_{t,a,n}, c_{t,a,n}) \leftarrow f_n(\mathbf{x}_{t,a})$
14:             Update leaf parameters:
                $\tilde{\mu}_{t,a,n} \leftarrow o_{t,a,n}, \quad \tilde{\sigma}_{t,a,n}^2 \leftarrow \frac{s_{t,a,n}^2}{c_{t,a,n}}$
15:             Increment arm parameters:
16:             $\tilde{\mu}_{t,a} \leftarrow \tilde{\mu}_{t,a} + \tilde{\mu}_{t,a,n}$
17:             $\tilde{\sigma}_{t,a}^2 \leftarrow \tilde{\sigma}_{t,a}^2 + \tilde{\sigma}_{t,a,n}^2$
18:             $c_{t,a} \leftarrow c_{t,a} + c_{t,a,n}$
19:         **end for**
20:         **if** bandit method is UCB **then**
21:             $U_{t,a} \leftarrow \tilde{\mu}_{t,a} + \sqrt{\nu^2 \tilde{\sigma}_{t,a}^2 \frac{\ln(t-1)}{c_{t,a}}}$
22:         **else if** bandit method is TS **then**
23:             $\tilde{r}_{t,a} \sim \mathcal{N}(\tilde{\mu}_{t,a}, \nu^2 \tilde{\sigma}_{t,a}^2)$
24:         **end if**
25:     **end for**
26:     **if** bandit method is UCB **then**
27:         $a_t \leftarrow \operatorname{argmax}_a U_{t,a}$
28:     **else if** bandit method is TS **then**
29:         $a_t \leftarrow \operatorname{argmax}_a \tilde{r}_{t,a}$
30:     **end if**
31:     Play $a_t$ and observe reward $r_{t,a_t}$
32: **end for**

---

final reward prediction. This should be accounted for in the total uncertainty as well, since it can otherwise be dominated by high uncertainty estimates from trees of low importance to the prediction. Accounting for the sample variances of the proposed tree contributions individually is a way of preventing such behavior.

### 3.4  Tree Ensemble Thompson Sampling

The way in which uncertainties are estimated by TEUCB can also be incorporated into Thompson Sampling (Thompson, 1933). In its traditional form, Thompson Sampling selects arms by sampling them from the posterior distribution describing each arm's probability of being optimal, given some (known or assumed) prior distribution and previously observed rewards. This can be achieved by sampling mean rewards from each arm's posterior distribution over expected rewards, and playing the arm with the largest sampled mean reward.

Using a TS-based approach, we propose to estimate the uncertainty in the predicted reward given an arm's context in the same way as in the case with UCB. However, in line 23 of Algorithm 1, instead of constructing confidence bounds, we sample mean rewards from the resulting distributions (here, interpreted as posterior distributions). Hence, the main difference from TEUCB is how the uncertainty is used to guide exploration. Due to its similarity with standard Thompson Sampling, we call this algorithm *Tree Ensemble Thompson Sampling* (TETS).

Regular Thomson Sampling is inherently a Bayesian approach. However, the framework may be extended to include the frequentist perspective as well, and the two views have been unified in a larger set of algorithms called *Generalized Thompson Sampling* (Li, 2013). It should be noted that TETS, as we present it here, is prior-free and should fall under the umbrella of Generalized TS. Although not the focus of this work, the algorithm could be modified to explicitly incorporate and utilize prior beliefs, making it Bayesian in the traditional sense.

### 3.5 Extension to Combinatorial Bandits

The framework outlined in Algorithm 1 is formulated to address the standard contextual MAB problem. However, TEUCB and TETS can easily be extended to the combinatorial semi-bandit setting. The main difference is in the way arms are selected, i.e., super arms instead of individual arms. In Algorithm 1, this corresponds to modifying lines 27 to

$$\mathcal{S}_t \leftarrow \mathrm{argmax}_{\mathcal{S}} \sum_{a \in \mathcal{S}} U_{t,a}, \tag{5}$$

and 29 to

$$\mathcal{S}_t \leftarrow \mathrm{argmax}_{\mathcal{S}} \sum_{a \in \mathcal{S}} \tilde{r}_{t,a}. \tag{6}$$

The selected super arm $\mathcal{S}_t$ at time $t$ is subsequently played in line 31. In this setting, the set of observed context-reward pairs would include all rewards received for each base arm in the selected super arms individually, which are generally more than one per time step.

### 3.6 Adaptation to XGBoost

Extreme gradient boosting (XGBoost) is a gradient boosting algorithm that utilizes ensembles of decision trees and techniques for computational efficiency (Chen & Guestrin, 2016). Gradient boosting creates new decision trees based on the gradient of the loss function. The ensemble output is based on the cumulative contribution of the decision trees within the ensemble.

When using XGBoost regression models, we can extract $o_n$ directly from the individual leaves in the constructed ensemble. The sample variance $s_n^2$ is not accessible directly, but we can easily calculate it. As a GBDT method, XGBoost calculates its outputs during the training procedure as the average difference between the target value and the value predicted from the collection of preceding trees in the ensemble, multiplied with a learning rate. Hence, the trees do not have predetermined weights as in, e.g., random forests. Instead, the relative size of the contributions of an assigned leaf to the final prediction depends on the particular paths that are traversed through the other trees, which may be different for every sample we observe in a leaf. Therefore, we cannot estimate $s_n^2$ from the variance in target values directly.

However, XGBoost is augmented with many useful features, one of them being staged predictions (XGBoost Developers, 2022b). This means that we can propagate the predicted outputs on sub-ensembles up to a certain tree on all data samples and cache these outputs. After recording these we include the next tree in the prediction as well, without having to start over. By utilizing this technique, $s_n^2$ is easily estimated from the previously observed samples assigned to a particular leaf. Furthermore, $c_n$ is simply the number of such samples. This gives us all the pieces of the puzzle needed to apply XGBoost to the TEUCB and TETS algorithms. In order to reduce bias when calculating $o_n$, $s_n^2$ and $c_n$, one may split the previously observed samples into two distinct data sets; one which is used for building the trees, and a second for the value estimations.

### 3.7 Adaptation to Random Forest

Random forest (Ho, 1995) is a supervised machine learning algorithm that utilizes decision trees as base learners together with bootstrapped aggregating (bagging). Bagging models randomly sample from the training data with replacement—reducing the variance of the model by providing the base learners with diverse subsets from the complete dataset. In addition to bagging, a random forest randomly samples a subset of the features (Hastie et al., 2009). For classification tasks, majority voting is commonly used to make predictions, while in regression, the average or weighted average of predictions is a common method.

Similar to XGBoost, random forests can also be incorporated into TEUCB and TETS. This is even less complicated in the case of random forests as the trees in a random forest do not depend on the predictions made by any of the other trees. Therefore, $s_n^2$ can be directly estimated from the variance in the target values, taking the tree's weight of $1/N$ into account. Hence, all the quantities of interest (i.e., $o_n$, $s_n^2$, and

$c_n$) are available from the trees of a random forest by looking at which observed samples are assigned to the terminal nodes.

As with XGBoost, one may consider dividing the previously observed samples into different subsets for fitting and value estimation. An alternative way for handling bias with random forests is to consider only the out-of-bag samples for computing $o_n$, $s_n^2$, and $c_n$. Random forests employ the concept of bagging—resampling with replacement—when building trees, which means roughly one-third of the samples will be omitted in the construction of any particular tree. Thus, they yield a natural way of providing an independent set of data samples for model evaluation without limiting the size of the training data set.

## 4 Experiments

In this section, we evaluate TEUCB and TETS and compare them against several well-known algorithms for solving the contextual bandit problem. For the implementations, we use the XGBoost library (Chen & Guestrin, 2016) to build gradient-boosted decision trees as our tree ensembles, as well as the version of random forests found in the scikit-learn library (Pedregosa et al., 2011). We first evaluate the algorithms on benchmark datasets. We then study them in the combinatorial contextual setting for solving the real-world problem of navigation over stochastic road networks in Luxembourg. The agent's cumulative regret is calculated by comparing the path to an oracle agent, who knows the expected travel duration for each edge given the current time of day. All the algorithms are evaluated w.r.t. cumulative regret averaged over ten random seeds and we report the average results. The setup of the experiments is inspired by and adapted from Russo et al. (2018).

### 4.1 Setup

We use the same procedure as Zhang et al. (2021) to transform a classification task into a contextual MAB problem. In each time step, the environment provides a context vector belonging to an unknown class. For the agent, each class corresponds to an arm, and the goal is to select the arm corresponding to the correct class. This results in a reward of 1, while an incorrect prediction results in a reward of 0.

Since only the context of a single (unknown) arm is provided, the agent must have a procedure for encoding it differently for each arm. This is generally done through positional encoding, where in the case of $K$ arms and $d$ data features, we form $K$ context vectors (each of dimensionality $Kd$) such that: $\mathbf{x}_1 = (\mathbf{x}; \mathbf{0}; \cdots; \mathbf{0})$, $\mathbf{x}_2 = (\mathbf{0}; \mathbf{x}; \cdots; \mathbf{0})$, $\cdots$, $\mathbf{x}_K = (\mathbf{0}; \mathbf{0}; \cdots; \mathbf{x})$. As no features are shared between any of the arms, this setup is an instance of a disjoint model, as opposed to a hybrid model (Li et al., 2010).

In the experiments in Section 4.2, we use the disjoint model for LinUCB, LinTS, NeuralUCB, and NeuralTS. However, due to the nature of discrete splitting by decision trees, we can encode the context more effectively using a hybrid model for tree-based methods. There, we simply append the corresponding labels as a single character for each arm respectively at the beginning of the context: $\mathbf{x}_k = (k, \mathbf{x})$, and mix numeric and categorical features in the context vectors. For LinUCB, LinTS, NeuralUCB, and NeuralTS, categorical features are one-hot-encoded.

In these experiments, we set the time horizon to 10,000 (except for the *Mushroom* dataset, where a horizon of 8,124 is sufficient to observe it entirely). For each of those time steps, the agents are presented with a feature vector $\mathbf{x}$ drawn randomly without replacement, which they encode for the different arms as described above. Subsequently, the agents predict the rewards for the individual arms according to the algorithms used. TEUCB and TETS (Algorithm 1) are described in Section 3. The TreeBootstrap (Elmachtoub et al., 2017), NeuralUCB (Zhou et al., 2020), NeuralTS (Zhang et al., 2021), LinUCB (Li et al., 2010) and LinTS (Agrawal & Goyal, 2013) algorithms are implemented according to their respective references.

### 4.2 Contextual Bandits

For this experiment, the data is collected from the UCI machine learning repository (Kelly et al., n.d.), where an overview of the used selected datasets is given in Table 1. *Magic* (Bock, 2007), which is short for *Magic Gamma Telescope*, contains only numerical features. The same is true for *Shuttle* (Statlog (Shuttle),

Table 1: Datasets overview.

| Dataset | Features | Instances | Classes |
|---------|----------|-----------|---------|
| Adult | 14 | 48,842 | 2 |
| Magic | 10 | 19,020 | 2 |
| Mushroom | 22 | 8,124 | 2 |
| Shuttle | 9 | 58,000 | 7 |

n.d.), also known as *Statlog*. All features of *Mushroom* (Mushroom, 1987) are categorical, and *Adult* (Becker & Kohavi, 1996) has a balanced distribution between numerical and categorical features.

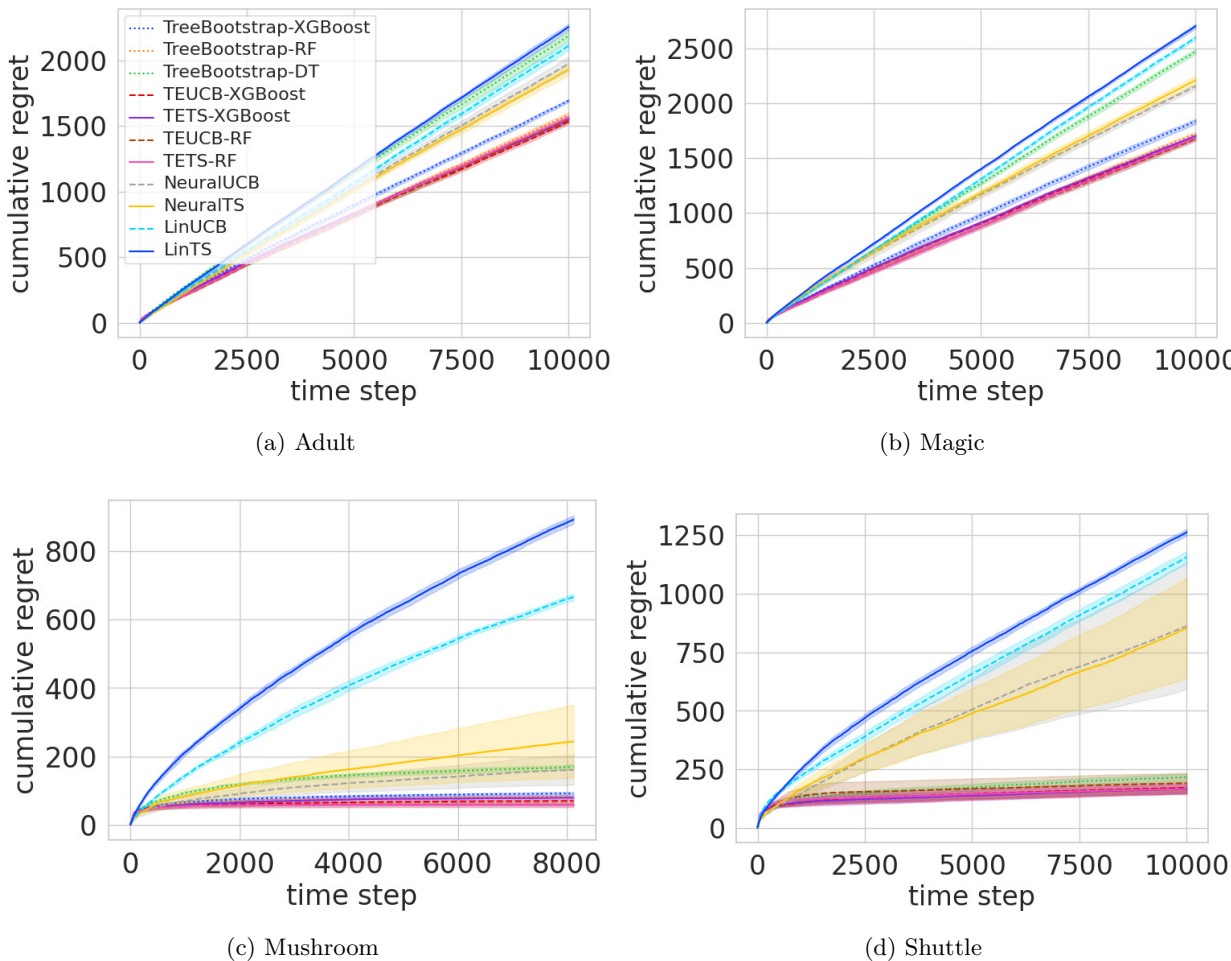

(a) Adult

(b) Magic

(c) Mushroom

(d) Shuttle

Figure 1: Comparison of contextual MAB algorithms on UCI datasets. Figures 1a, 1b, 1c, and 1d share the same color scheme for consistency, but the legend is only presented in 1a for improved visibility.

### 4.2.1 Implementation

For all datasets in Table 1, the neural network agents use a network architecture of one hidden layer with 100 neurons. Regarding NeuralUCB, NeuralTS, LinUCB, and LinTS, we search the same sets of hyperparameters as described by Zhang et al. (2021), who run these agents on the same datasets, and select the parameters with the best performance. One difference in our experiments is that we added a dropout probability of 0.2 when training the neural networks, which had a positive impact on the performance. Furthermore, each of the networks is trained for 10 epochs. For all tree ensemble bandits, we use XGBoost and random forest regressors with MSE loss and ensembles of 100 trees.

Table 2: Average regret accumulated by agents after the final step, with standard deviation, and number of hours required to run one experiment with CPU. Runtime in hours on GPU is also reported where it applies.

| | | Adult | Magic | Mushroom | Shuttle |
|---|---|---|---|---|---|
| TEUCB-XGBoost | Mean ± SD | **__1532.5__** ± 27.8 | 1685.7 ± 27.7 | 69.2 ± 8.6 | 171.4 ± 40.0 |
| | CPU | 0.13 | 0.14 | 0.09 | 0.08 |
| TETS-XGBoost | Mean ± SD | 1548.0 ± 30.9 | 1703.2 ± 25.2 | 78.6 ± 11.8 | 165.7 ± 31.2 |
| | CPU | 0.13 | 0.14 | 0.09 | 0.08 |
| TEUCB-RF | Mean ± SD | 1550.6 ± 32.9 | **__1678.5__** ± 31.3 | 58.2 ± 10.9 | 190.8 ± 68.9 |
| | CPU | 0.12 | 0.13 | 0.10 | 0.12 |
| TETS-RF | Mean ± SD | 1566.3 ± 34.2 | 1688.6 ± 38.3 | **__57.7__** ± 8.1 | 166.3 ± 37.1 |
| | CPU | 0.12 | 0.13 | 0.10 | 0.12 |
| NeuralUCB | Mean ± SD | 1974.0 ± 82.0 | 2155.2 ± 39.5 | 160.8 ± 68.9 | 862.4 ± 428.8 |
| | CPU | 10 | 10 | 7.4 | 8.1 |
| | GPU | 1.2 | 0.68 | 0.37 | 1.4 |
| NeuralTS | Mean ± SD | 1929.7 ± 68.3 | 2209.5 ± 62.2 | 243.4 ± 168.9 | 853.8 ± 340.3 |
| | CPU | 10 | 10 | 7.4 | 8.1 |
| | GPU | 1.2 | 0.68 | 0.37 | 1.4 |
| LinUCB | Mean ± SD | 2109.8 ± 51.8 | 2598.4 ± 25.4 | 666.2 ± 14.7 | 1156.3 ± 41.3 |
| | CPU | 0.02 | 0.02 | 0.04 | 0.11 |
| LinTS | Mean ± SD | 2253.2 ± 37.7 | 2703.8 ± 30.4 | 892.2 ± 18.6 | 1263.0 ± 22.1 |
| | CPU | 0.02 | 0.02 | 0.04 | 0.11 |
| TreeBootstrap-XGBoost | Mean ± SD | 1693.2 ± 18.8 | 1836.1 ± 36.7 | 91.6 ± 5.6 | 164.1 ± 24.4 |
| | CPU | 0.97 | 1.2 | 0.10 | 0.27 |
| TreeBootstrap-RF | Mean ± SD | 1584.3 ± 32.5 | 1722.9 ± 28.0 | 82.5 ± 3.3 | **__161.9__** ± 29.2 |
| | CPU | 3.8 | 4.2 | 1.8 | 4.2 |
| TreeBootstrap-DT | Mean ± SD | 2187.2 ± 57.1 | 2473.3 ± 35.3 | 168.9 ± 11.0 | 216.0 ± 22.9 |
| | CPU | 0.08 | 0.09 | 0.04 | 0.08 |

Initial test runs revealed that the agents are robust w.r.t. different choices of hyper-parameters for XGBoost (XGBoost Developers, 2022a), and consequently, we settle on the default values. The only exception is the maximum tree depth, which we set to 10. The same maximum tree depth is used for the implementation of random forests. The number of initial random arm selections $T_I$ is set to 10 times the number of arms for all tree-based algorithms, and the exploration factor of TEUCB and TETS is set to $\nu = 1$. As for the decision tree (DT) algorithm used for TreeBootstrap, we use the scikit-learn library to employ CART (Breiman et al., 2017), which is free of tunable parameters. This is similar to the implementation presented in the original work on TreeBootstrap (Elmachtoub et al., 2017).

Another point we noted during our initial test runs was that accurate predictions seem to be more important than less biased estimates obtained by splitting the previously observed samples into distinct data sets for ensemble fitting and value calculations respectively. Therefore, we use all observed context-reward pairs for both purposes in all experiments with TEUCB and TETS.

In order to avoid unnecessary computations and speed up the runs, TEUCB and TETS do not build new tree ensembles from scratch at each time step. Instead, they only consider which leaves the latest observation assigned in each tree and update the corresponding $o_n$, $s_n^2$, and $c_n$ parameters. Initially, when newly observed samples may have a relatively large effect on the optimal tree ensemble, re-building happens more frequently. However, as the effect that the samples are expected to have on the tree architectures degrades over time, less re-building takes place. More precisely, re-building happens when the function $\lceil 8 \ln(t) \rceil$ increases by one compared to the previous time step.

The experimental results are obtained using an NVIDIA A40 GPU for NeuralUCB and NeuralTS, and a desktop CPU for the other agents. To illustrate the differences in computational performance between the agents, we report the runtime of the neural agents on the same CPU as well.

### 4.2.2 Experimental Results

The results of the experiments described above are presented in Fig. 1 and Table 2. We observe that the tree ensemble methods consistently outperform all other models by a large margin, both with XGBoost and

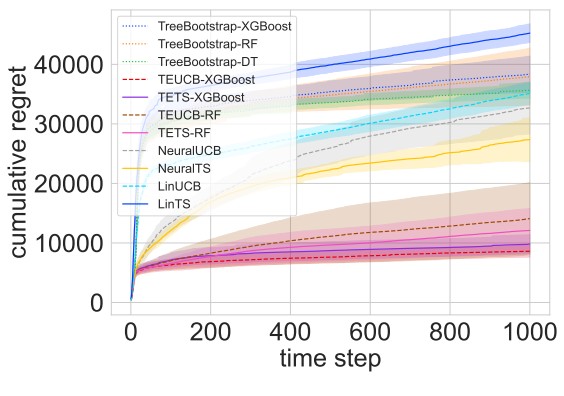 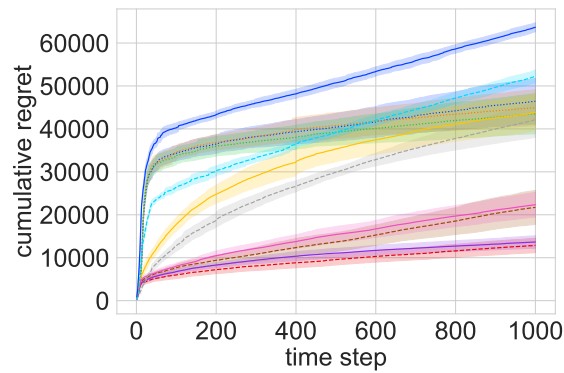

(a) Cumulative regret on paths problem instance 1 (b) Cumulative regret on paths problem instance 2

Figure 2: Experimental results on real-world road network navigation in Luxembourg.

random forest, and yield significantly lower cumulative regrets. Furthermore, TEUCB and TETS tend to perform better than TreeBootstrap on the adult, magic, and mushroom data sets.

On the shuttle data set, with seven different arms, the TreeBootstrap agents exhibit comparable and even slightly better average performance in terms of regret minimization. It appears that TreeBootstrap's assigning of a separate tree-based model for each arm is beneficial for this problem. However, this comes at the expense of having to fit multiple models in each time step, which is time-consuming and computationally demanding. Furthermore, its inability to generalize the reward predictions of distinct but related arms may in some cases be disadvantageous, especially for larger arm sets. Comparing TEUCB vs. TETS, and XGBoost vs. random forest (i.e., the different design choices within our framework), there is no clear winner as they all tend to perform comparably (and very effectively) on different data sets. In addition to regret minimization, the CPU experiments indicate that TEUCB and TETS are significantly more efficient than their neural counterparts from a computational perspective. Notably, LinUCB, LinTS, and TreeBootstrap-DT are the most efficient methods in terms of computation. However, they do not minimize regret as effectively as most other agents in our data sets.

### 4.3 Combinatorial Contextual Bandits

In this section, we investigate the combinatorial contextual bandit methods on a real-world application, where we study two scenarios corresponding to performing the most efficient navigation over the real-world road network of Luxembourg. This problem is crucial with the emergence of electric vehicles to mitigate the so-called range anxiety. Similar navigation problems have recently been studied from a CMAB perspective, but often without contextual information (Åkerblom et al., 2023) or limited to neural bandit methods (Hoseini et al., 2022). CMAB methods are well-suited to the navigation problem since the traversal time of each road segment can be highly stochastic and dependent on local factors (e.g., road works, traffic congestion, stop lights) about which knowledge may be gathered through sequential interactions with the environment.

#### 4.3.1 Implementation

We model the road network of Luxembourg via a graph $\mathcal{G}(\mathcal{V}, \mathcal{E})$, with $|\mathcal{V}| = 2,247$ vertices and $|\mathcal{E}| = 5,651$ edges. The vertices represent intersections in the road network, and the edges represent individual road segments connecting the intersections. In this scenario, edges correspond to base arms, and paths (i.e., ordered sequences of edges) correspond to super arms. Each vertex has a coordinate consisting of longitude, latitude, and altitude values. For all edges $e \in \mathcal{E}$, the contextual vector $\mathbf{x}_e$ describes each road segment in the network. The agent is presented with a vector containing contextual data according to Table 3.

Edge traversal times have been collected using the Luxembourg SUMO Traffic (LuST) simulation scenario (Codeca et al., 2015). The recorded edge traversal times are used to form kernel density estimators (KDE) (Weglarczyk, 2018) for each edge. If an edge does not contain any recorded traversals, the expected traversal

Table 3: The variables included in the contextual vector describing each edge in the graph.

| Variable | Description |
| --- | --- |
| $x$ | Start position along x-axis. |
| $y$ | Start position along y-axis. |
| $z$ | Start position along z-axis. |
| $x'$ | End position along x-axis. |
| $y'$ | End position along y-axis |
| $z'$ | End position along z-axis |
| $\sqrt{(x-x')^2}$ | Euclidean distance along x-axis. |
| $\sqrt{(y-y')^2}$ | Euclidean distance along y-axis. |
| $\sqrt{(z-z')^2}$ | Euclidean distance along z-axis. |
| speed_limit | Maximum speed limit |
| stop | A boolean if edge includes a stop |
| time | The current time of day |

time is set to the length of the edge divided by the speed limit. At each time step $t$, the time of day is randomly sampled, and used for updating the expected travel times of all edges and the corresponding KDE's. We generate edge-specific feedback by individually sampling travel times from the KDE of each edge on the chosen path.

As the graph contains more than 5,000 edges, the agents need to learn how the contextual features impact the expected travel time. The road types in the graph are highways, arterial roads, and residential streets, with a total length of 955 km. In our experiments, we specifically study two problem instances (characterized by different start and end nodes), referred to as problem instance 1 and problem instance 2. The paths selected by the different agents during a single run are visualized for problem instance 1 in Fig. 3 (in the Appendix).

An agent predicts the expected travel time for each of the edges in the graph, and then solves the shortest path problem using Dijkstra's algorithm (Dijkstra, 1959). The agents are evaluated based on the sum of the expected travel times for all edges forming the traversed path compared to the expected travel time of the optimal path $\mathcal{S}^*$.

For this experiment, the neural agents utilize a neural network with two hidden layers, both containing 100 fully connected neurons. A dropout probability of 0.2 is used and after a parameter search over the same sets of values as in Section 4.2.1, we set $\lambda = 0.1$ and $\nu = 0.001$. Furthermore, the network is trained over 10 epochs with the option of early stopping if the MSE loss does not tend to keep improving. We implement the tree ensemble bandits utilizing XGBoost and random forest regressors with maximum tree depths of 10; all other hyper-parameters being set to their default values. The number of trees is set to $N = 100$ for all agents with tree ensembles, and the initial random selection is set to $T_I = 10$ paths. An exploration factor of $\nu = 1$ is used for TEUCB and TETS and the frequency of re-building their tree ensembles is the same as described in Section 4.2.1.

### 4.3.2 Experimental Results

Fig. 2 shows the results of the TEUCB and TETS methods along with the baselines. The road network and the agents' traversals are presented in Fig. 3 (in the Appendix). The frequency by which an edge has been traversed is indicated by the red saturation, where more traversals correspond to a higher saturation. It is worth noting that LinUCB, LinTS, and the TreeBootstrap require excessive exploration as their models are edge-specific. In contrast, TETS, TEUCB, NeuralTS, and NeuralUCB train one model with the possibility of generalizing the expected travel time predictions over different edges. We observe that using both XGBoost and random forest, TEUCB and TETS significantly outperform the other methods. In this setting, if an agent can generalize well what it has learned from one arm's reward distribution to the other arms, it can then effectively avoid over-exploration, as indicated by both the regret plots in Fig. 2 and the agent's specific path selections in Fig. 3. Comparing the two different tree ensemble methods, XGBoost seems to perform better on the navigation task compared to random forests in the TEUCB and TETS frameworks.

The neural methods tend to outperform LinUCB and LinTS. Compared to TreeBootstrap, however, the advantage is less clear, and seems to depend on the particular time horizon $T$. NeuralUCB and NeuralTS

tend to learn faster, which is unsurprising since they can generalize reward predictions over different specific arms. After an initial period of heavy exploration, however, the TreeBootstrap agents appear to make good path selections more frequently, and yield cumulative reward curves that are more similar to those of TEUCB and TETS in slope.

All the agents were run on a single desktop CPU, apart from NeuralUCB and NeuralTS, which were run on an NVIDIA A40 GPU. NeuralTS and NeuralUCB tend to require highly parameterized networks and substantial hyper-parameter tuning to achieve deliberate exploration, which can make them time-consuming while running and searching for good parameter values. In terms of runtime, the experiments took about an hour on the GPU for a neural agent. It is also noticeable that the neural agents appear to be sensitive to the weight initialization of the networks, which leads to higher variance (see Fig. 2). In contrast, TETS and TEUCB achieve solid results for a large range of parameter settings, yield lower regret, and tend to run at comparable speed on a single CPU (instead of GPU). The experiments with TEUCB and TETS took about 1.5 hours with XGBoost, and 5 hours using random forests as the tree ensemble methods. The runtime of TreeBootstrap also tends to depend heavily on the particular tree model used. With a single decision tree per arm, the experiments took 0.5 hours, which is about the same as LinUCB and LinTS. Using tree ensembles, they took about 4 and 20 hours with XGBoost and random forests, respectively.

## 5   Conclusion

We developed a novel framework for contextual multi-armed bandits using tree ensembles. Within this framework, we adapted the two commonly used methods for handling the exploration-exploitation dilemma: UCB and Thompson Sampling. Furthermore, we extended the framework to handle combinatorial contextual bandits, enabling more complex action selection at each time step. To demonstrate the effectiveness of the framework, we conducted experiments on benchmark datasets using the XGBoost and random forest tree ensemble methods. Additionally, we employed it for navigation over stochastic real-world road networks, modeled as a combinatorial contextual bandit problem.

Across all problem instances, the introduced tree ensemble-based methods, TEUCB and TETS, consistently demonstrated effectiveness in minimizing regret while requiring relatively low computational resources. In many cases, these methods significantly outperformed neural network-based methods that are often considered state-of-the-art for contextual bandits, in terms of both accuracy and computational cost. Hence, this work indicates that using tree ensembles offers several advantages over other machine-learning models for these problems. Furthermore, compared to other tree-based methods, TEUCB and TETS exhibited similar or better capabilities in minimizing regret while maintaining computational efficiency, even for problems with large numbers of arms. These results indicate that using a single tree ensemble to predict the reward distributions of all arms, particularly as implemented in TEUCB and TETS, facilitates effective generalization between arms based on context, enabling more efficient learning with fewer samples.

The study primarily focuses on the practical applicability of tree ensembles for bandits in various settings. Therefore, we did not provide a theoretical analysis of the respective regrets. For future work, exploring theoretical regret bounds for the proposed tree ensemble methods could offer deeper insights and a better understanding of the methods presented in this work.

## Acknowledgments

The computations and data handling were enabled by resources provided by the National Academic Infrastructure for Supercomputing in Sweden (NAISS), partially funded by the Swedish Research Council through grant agreement no. 2022-06725. The work of Niklas Åkerblom was partially funded by the Strategic Vehicle Research and Innovation Programme (FFI) of Sweden, through the project EENE (reference number: 2018-01937).

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

# A Appendix

## A.1 Nomenclature

Table 4: Table of notations used in this paper.

| Notation | Description |
|---|---|
| $a$ | Arm/action |
| $a^*$ | Optimal arm |
| $a_t$ | Arm selected at time step $t$ |
| $\mathcal{A}$ | Action space, set of all arms |
| $c_n$ | Number of training samples assigned to a leaf in regression tree $n$ |
| $f$ | Tree ensemble regressor consisting of $N$ regression trees fitted in unison |
| $f_n$ | $n$th individual tree of the ensemble |
| $m_{t,a}$ | Number of times arm $a$ has been selected up to time step $t$ |
| $\mathcal{N}(\mu, \sigma^2)$ | Normal distribution with mean $\mu$ and variance $\sigma^2$ |
| $o_n$ | Output value associated with a leaf in regression tree $n$ |
| $q$ | Function of expected reward |
| $r$ | Observed reward |
| $\tilde{r}$ | Estimated reward |
| $R$ | Regret |
| $s_n^2$ | Sample variance of the output value associated with a leaf in regression tree $n$ |
| $\mathcal{S}$ | Super arm, a set of arms |
| $\mathcal{S}^*$ | Optimal super arm |
| $\mathcal{S}_t$ | super arm selected at time step $t$ |
| $t$ | time step |
| $U$ | Upper confidence bound |
| $\mathbf{x}$ | Context/feature vector |
| $\mu$ | True mean |
| $\tilde{\mu}$ | Estimated mean |
| $\nu$ | Exploration factor |
| $\sigma^2$ | True variance |
| $\tilde{\sigma}^2$ | Estimated variance |
| $\{x_i\}_{i=1}^I$ | List of $I$ entries |

## A.2 Further Elaboration on the Independence Assumption

Given a tree ensemble $f = \{f_1, f_2, \ldots, f_N\}$, the prediction for a context vector $\mathbf{x}$ is the sum of individual tree outputs:

$$p(\mathbf{x}) = \sum_{i=1}^{N} o_i(\mathbf{x}), \tag{7}$$

where each $o_i(\mathbf{x})$ is a random variable representing the output of tree $i$ when the input is $\mathbf{x}$. Assuming the output of each tree $o_i(\mathbf{x})$ has a mean $\mu_i$ and a variance $\frac{\sigma_i^2}{c_i}$ (where $c_i$ is the number of training samples assigned to the leaf of tree $i$ that $\mathbf{x}$ ends up in), the variance of the ensemble's prediction is:

$$\text{Var}(p(\mathbf{x})) = \text{Var}\left(\sum_{i=1}^{N} o_i(\mathbf{x})\right). \tag{8}$$

Under the *independence assumption* between the trees, the variance of the sum is simply the sum of the variances of the individual trees:

$$\text{Var}(p(\mathbf{x})) = \sum_{i=1}^{N} \text{Var}(o_i(\mathbf{x})) = \sum_{i=1}^{N} \frac{\sigma_i^2}{c_i}. \tag{9}$$

This expression allows us to estimate the total uncertainty in the ensemble's prediction by summing up the individual uncertainties of each tree.

In reality, especially in boosting algorithms, there might be some correlation between the trees. Let us denote the correlation between the outputs of two trees $i$ and $j$ as $\rho_{ij}$. The variance of the sum of tree outputs in the presence of correlations becomes:

$$\text{Var}(p(\mathbf{x})) = \sum_{i=1}^{N} \frac{\sigma_i^2}{c_i} + 2 \sum_{i=1}^{N-1} \sum_{j=i+1}^{N} \rho_{ij} \sqrt{\frac{\sigma_i^2}{c_i} \cdot \frac{\sigma_j^2}{c_j}}. \tag{10}$$

Here, $\rho_{ij}$ represents the correlation coefficient between the outputs of trees $i$ and $j$. If the correlations $\rho_{ij}$ are positive, they increase the total variance of the ensemble's prediction. However, if we assume independence ($\rho_{ij} = 0$), the second term vanishes, simplifying the variance to just the sum of individual variances, i.e.,

$$\text{Var}(p(\mathbf{x})) \approx \sum_{i=1}^{N} \frac{\sigma_i^2}{c_i}. \tag{11}$$

In the context of bagging-based ensembles like random forests, where trees are trained on different subsets of data and with random feature selection, the correlations $\rho_{ij}$ are typically small, making this approximation reasonable. For boosting-based methods, which sequentially fit trees, the correlations might be stronger, implying that our approximation could be less accurate.

However, we argue that this approximation often yields a lower bound for the actual variance, where this lower bound provides useful insights into the ensemble's uncertainty, offering a simplified way to quantify prediction variability. For this, we argue that negative correlations are less common than positive ones in ensemble learning primarily due to the nature of how individual models are trained and how ensemble methods perform.

In tree ensemble methods, individual models (e.g., decision trees) are trained on subsets of the same data and tend to capture underlying patterns that are common across these subsets. This shared learning process often leads to similar predictions, resulting in positive correlations. In boosting, in particular, trees are added sequentially, with each new tree focused on correcting the errors (residuals) of the previous ensemble. As the boosting process continues, new trees are highly dependent on the earlier trees' mistakes. Although boosting can sometimes create predictions that 'pull' in opposite directions (implying a negative correlation locally), the overall direction of learning is typically aligned to reduce residual error. This alignment contributes to positive correlations because all trees are ultimately working together to approximate the same target function. Therefore, we can assume that in boosting, the overall correlation is positive. On the other hand, as mentioned, bagging methods such as random forests use randomness (bootstrapping data samples and feature selection) to reduce the correlation between individual trees (i.e., the correlations can be discarded).

In summary, we expect the independence assumption between the trees (i.e., $\rho_{ij} = 0$) to hold reasonably well for bagging methods. On the other hand, this assumption usually leads to an underestimate of the total variance for boosting methods. This means that our calculated variance: $\sum_{i=1}^{N} \frac{\sigma_i^2}{c_i}$ serves as a *lower bound* on the true variance. Interestingly, some bandit algorithms, such as UCB, are known to suffer from *over-exploration* (Russo & Roy, 2014), and thus underestimating the variance could even be helpful to mitigate this issue and lead to more efficient learning.

### A.3 Calculating Leaf Values

The code blocks presented in this section outline the procedure by which the leaf values $o_{n,l}$, $s_{n,l}^2$ and $c_{n,l}$ for a leaf $l$ in tree $n$ are obtained for the implementation of TEUCB and TETS with XGBoost and random forests, respectively.

---

**Algorithm 2** Set Leaf Values - XGBoost

---

1: **Input:** Tree ensemble regressor $f$, number of trees in ensemble $N$, base predictor value $b$, learning rate $\eta$, set of $D$ training data points $\{(\mathbf{x}_i, r_i)\}_{i=1}^D$.
2: $o_{n,l}, s_{n,l}, c_{n,l} = 0 \quad \forall$ trees $n$, leaves $l$
3: **for** $i = 1$ **to** $D$ **do**
4:     **for** $n = 1$ **to** $N$ **do**
5:         Check which leaf $l$ that the context $\mathbf{x}_i$ is assigned to in tree $n$
6:         $c_{n,l} = c_{n,l} + 1$
7:         Get prediction on $\mathbf{x}_i$ from previous trees: $p_{n-1} = b + \sum_{j=1}^{n-1} f_j(\mathbf{x}_i)$
8:         $o_{n,l,c_{n,l}} = \eta(r_i - p_{n-1})$
9:     **end for**
10: **end for**
11: **for** $n = 1$ **to** $N$ **do**
12:     **for** leaves $l$ in tree $n$ **do**
13:         $o_{n,l} = \frac{1}{c_{n,l}} \sum_{i=1}^{c_{n,l}} o_{n,l,i}$
14:         $s_{n,l}^2 = \frac{1}{c_{n,l}-1} \sum_{i=1}^{c_{n,l}} (o_{n,l,i} - o_{n,l})^2$
15:     **end for**
16: **end for**

---

**Algorithm 3** Set Leaf Values - Random Forest

---

1: **Input:** Tree ensemble regressor $f$, number of trees in ensemble $N$, set of $D$ training data points $\{(\mathbf{x}_i, r_i)\}_{i=1}^D$.
2: $o_{n,l}, s_{n,l}, c_{n,l} = 0 \quad \forall$ trees $n$, leaves $l$
3: **for** $i = 1$ **to** $D$ **do**
4:     **for** $n = 1$ **to** $N$ **do**
5:         Check which leaf $l$ that the context $\mathbf{x}_i$ is assigned to in tree $n$
6:         $c_{n,l} = c_{n,l} + 1$
7:         $o_{n,l,c_{n,l}} = f_j(\mathbf{x}_i)$
8:     **end for**
9: **end for**
10: **for** $n = 1$ **to** $N$ **do**
11:     **for** leaves $l$ in tree $n$ **do**
12:         $o_{n,l} = \frac{1}{c_{n,l}} \sum_{i=1}^{c_{n,l}} o_{n,l,i}$
13:         $s_{n,l}^2 = \frac{1}{c_{n,l}-1} \sum_{i=1}^{c_{n,l}} (o_{n,l,i} - o_{n,l})$
14:     **end for**
15: **end for**

---

### A.4 Path Selections

In this section, we present maps of the routes selected by the respective agents throughout one run of the navigation task.

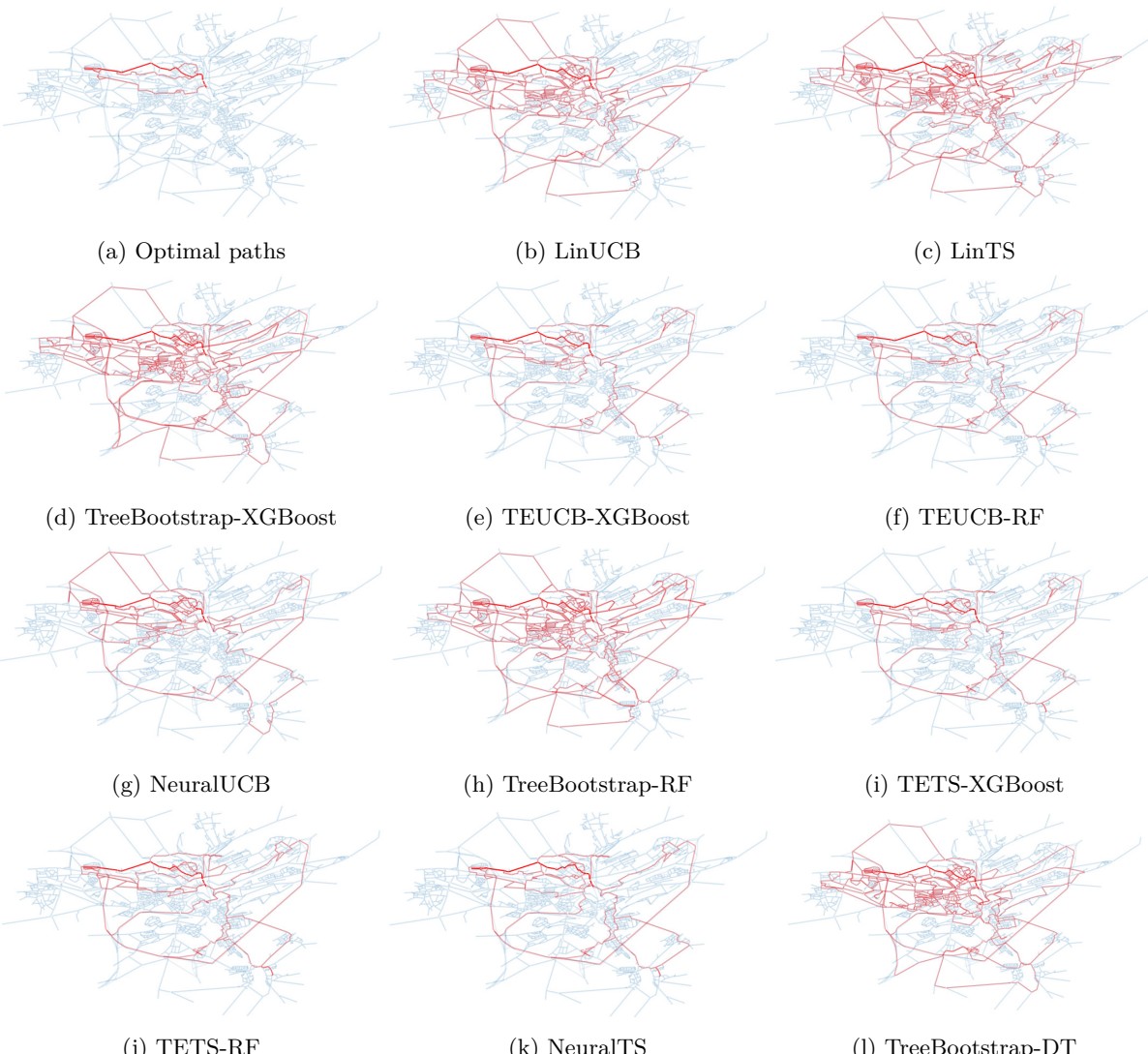

Figure 3: The road network of Luxembourg shows the trajectories of different agents for the experiment on problem instance 1, where the corresponding cumulative regret is presented in Fig. 2a. The plots show all the paths selected by the agents during a full run of the experiment, where a higher level of opacity indicates that a road segment was more frequently part of a traveled path.

### A.5 Sensitivity Analysis

Here, we investigate the robustness of TEUCB and TETS to changes in a number of key (hyper)parameters. To do so, we run the same experiments as presented in Section 4.2, but varying one parameter at a time. In addition, we study how robust the methods are to delays in reward feedback, and compare the performance of TEUCB and TETS to TreeBootstrap.

### A.5.1 Delayed Feedback

As done by Zhang et al. (2021), in the experiments with delayed feedback presented in Fig. 4, the rewards associated with an agent's actions are fed in batches of varying batch size, rather than one by one instantly after the action is taken. This setting appears to occur naturally in several real-world applications of multi-armed bandits, as discussed by Chapelle & Li (2011). Their findings suggest that Thompson Sampling scales better with delay in rewards than UCB methods, which is also supported by the experiments of Zhang et al. (2021). It is not clear from our experiments whether this is the case for TEUCB, TETS, and TreeBootstrap (which, as discussed in Section 1.1, can be viewed as a Thompson Sampling method). Comparing with the results of Zhang et al. (2021) for NeuralUCB and NeuralTS on the same datasets, however, it appears that the tree-based methods we study are more resilient to delays in the rewards.

### A.5.2 Varied Tree Depth

In Fig. 5, we study the robustness of TEUCB and TETS towards different depths for the regression trees in the ensembles. We observe that a significant change in tree depth may influence the performance of TEUCB and TETS when used with both XGBoost and random forests. We note that random forests tend to benefit from deeper trees, while XGBoost performs better with shallower ones. This is particularly interesting as the depth is only a maximum value for XGBoost, and the algorithm is designed to automatically prune trees where deemed beneficial for the reduction of overfitting. The extent to which this is utilized can be controlled by changing the XGBoost parameters $\gamma$ and $\lambda$, though we use the default values. As a final note on the effect of tree depth, we observe that a depth of 10, which is used in the comparison study in Section 4.2, is suitable.

### A.5.3 Varied Ensemble Size

In Fig. 6 we study how the performance of TEUCB and TETS varies with different numbers of trees making up the tree ensembles used to predict rewards. In this study, we see that initially going from a relatively small ensemble of trees to a larger one can greatly improve the performance, which makes sense. However, after this initial setup, the results stay stable when increasing the number of trees further. XGBoost appears to be particularly sensitive to selecting too few trees. This issue may potentially be mitigated by adjusting the learning rate parameter $\eta$, though we have not investigated this. Further, we note that setting the number of trees to 100, as in the comparison with other bandit methods in Section 4.2, is shown to be a good choice.

### A.5.4 Varied Exploration Factor

Finally, in Fig. 7 we investigate the robustness of TEUCB and TETS to different exploration factors used to control the level of exploration. The results demonstrate that the methods are robust to changes in the exploration factor within a wide range of values. Only one observation deviates significantly from the pattern, in which an exploration factor of 100 is used for TEUCB with XGBoost. For both datasets, this value results in a significantly larger average cumulative regret compared to all other evaluated methods and parameter values. For all other values and agents, the results are stable and consistent.

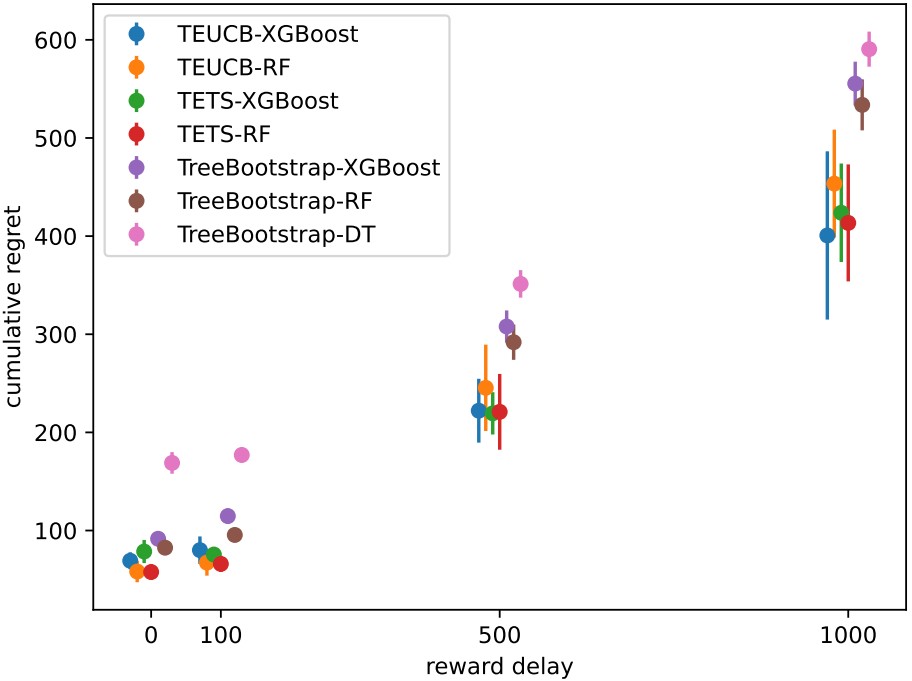

(a) Cumulative regret with different levels of reward delays on the mushroom dataset

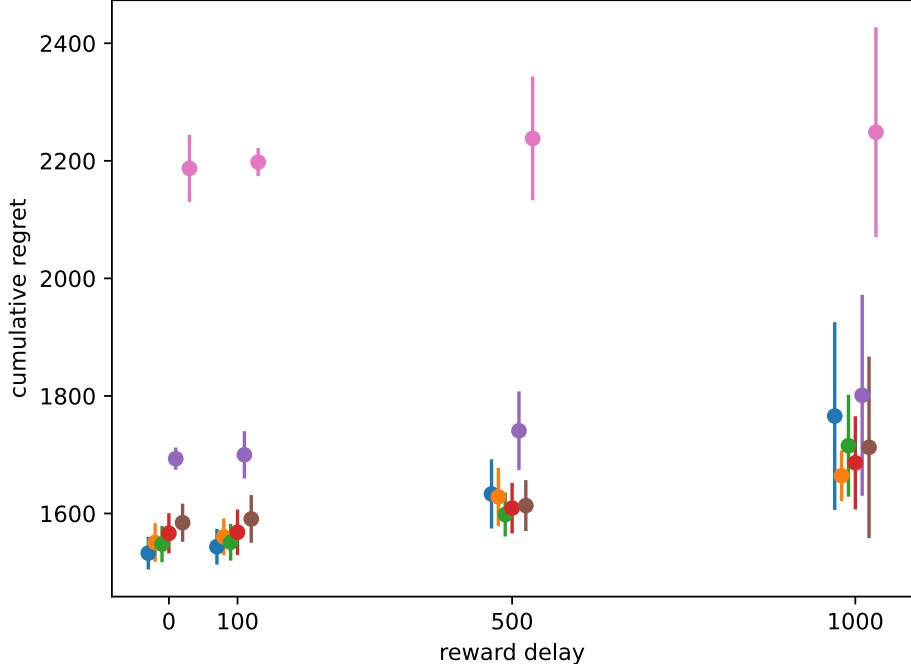

(b) Cumulative regret with different levels of reward delays on the adult dataset

Figure 4: Comparison of TEUCB, TETS, and TreeBootstrap on the mushroom and the adult datasets with different levels of delays. Cumulative regret for a single experiment is calculated over 10,000 time steps and repeated 10 times. The results display the average cumulative regret plus/minus the standard deviation for each respective agent. Hyperparameters are selected as in Section 4.2. All agents are evaluated on the same levels of reward delays, but each agent is shifted slightly horizontally for visualization purposes.

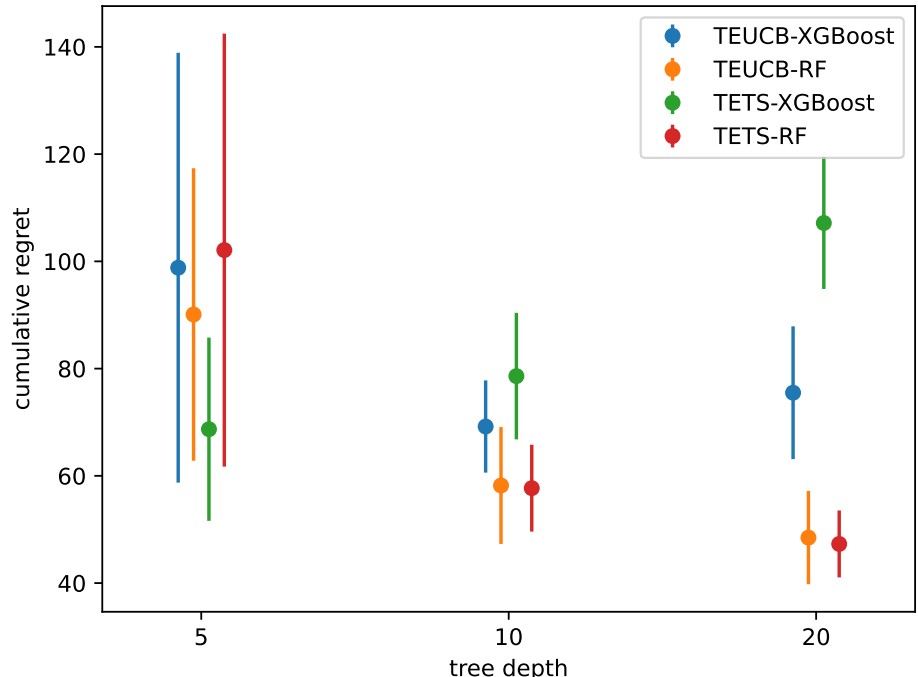

(a) Cumulative regret with varying tree depths on the mushroom dataset

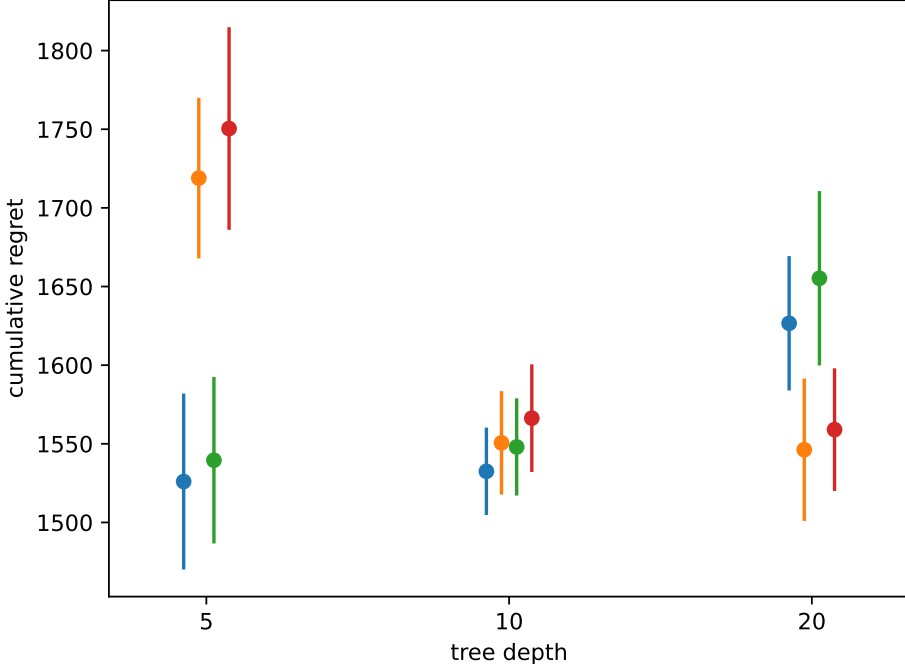

(b) Cumulative regret with varying tree depths on the adult dataset

Figure 5: Comparison of TEUCB and TETS on the mushroom and the adult datasets with different tree depths. Cumulative regret for a single experiment is calculated over 10,000 time steps and repeated 10 times. The results display the average cumulative regret plus/minus the standard deviation for each respective agent. Hyperparameters other than tree depth are selected as in Section 4.2. All agents are evaluated with the same tree depths, but each agent is shifted slightly horizontally for visualization purposes.

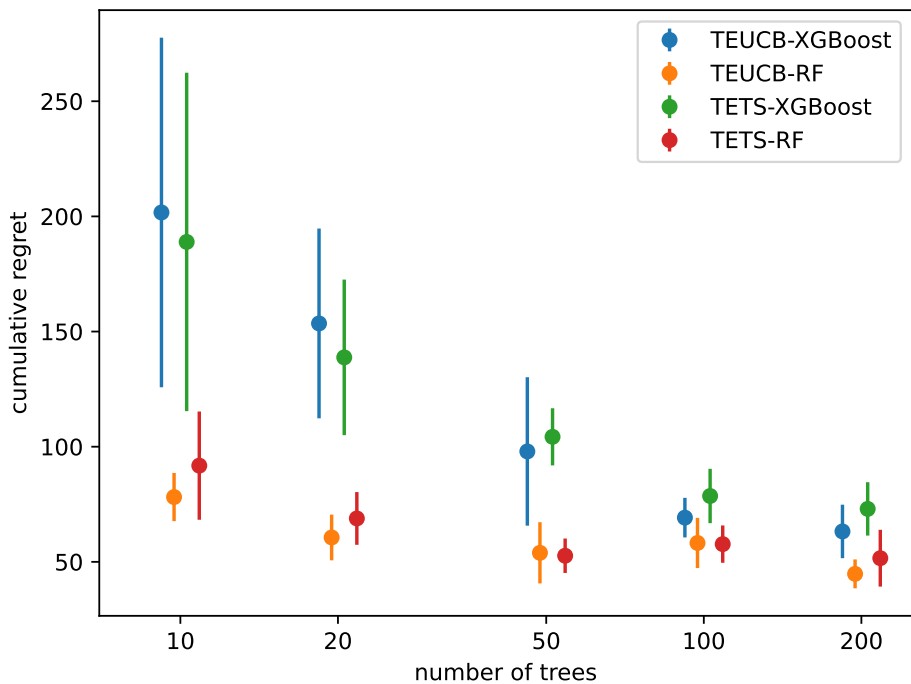

(a) Cumulative regret with varying ensemble sizes on the mushroom dataset

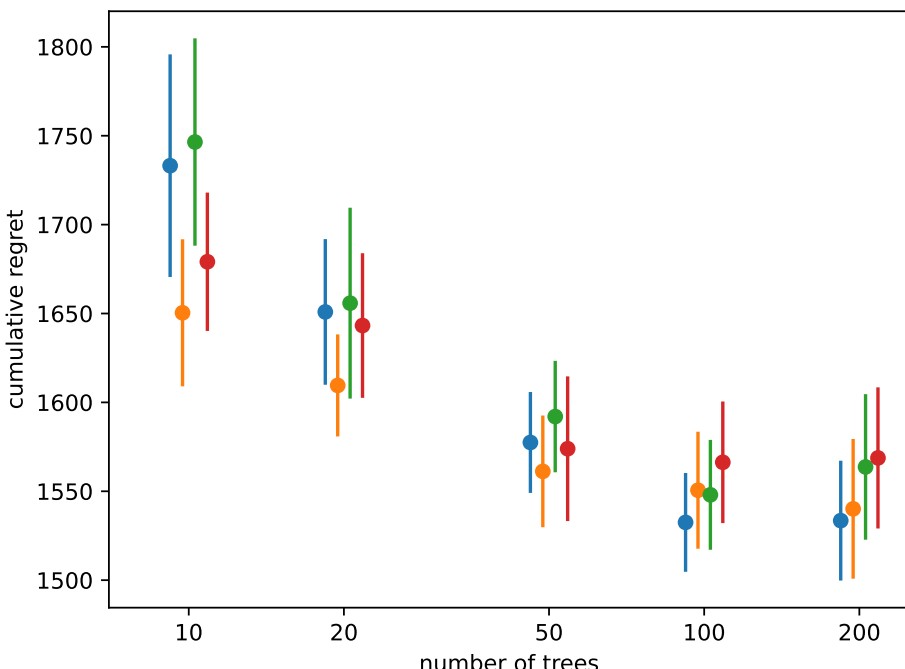

(b) Cumulative regret with varying ensemble sizes on the adult dataset

Figure 6: Comparison of TEUCB and TETS on the mushroom and the adult datasets with different numbers of trees in the ensembles. Cumulative regret for a single experiment is calculated over 10,000 time steps and repeated 10 times. The results display the average cumulative regret plus/minus the standard deviation for each respective agent. Hyperparameters other than the numbers of trees are selected as in Section 4.2. All agents are evaluated with the same number of trees, but each agent is shifted slightly horizontally for visualization purposes.

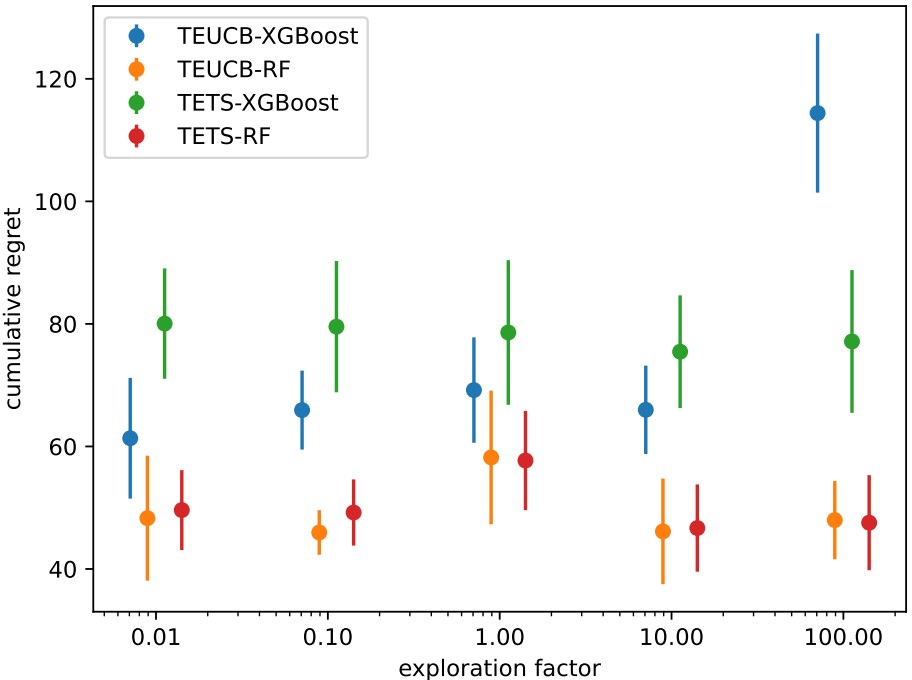

(a) Cumulative regret with varying exploration factors on the mushroom dataset

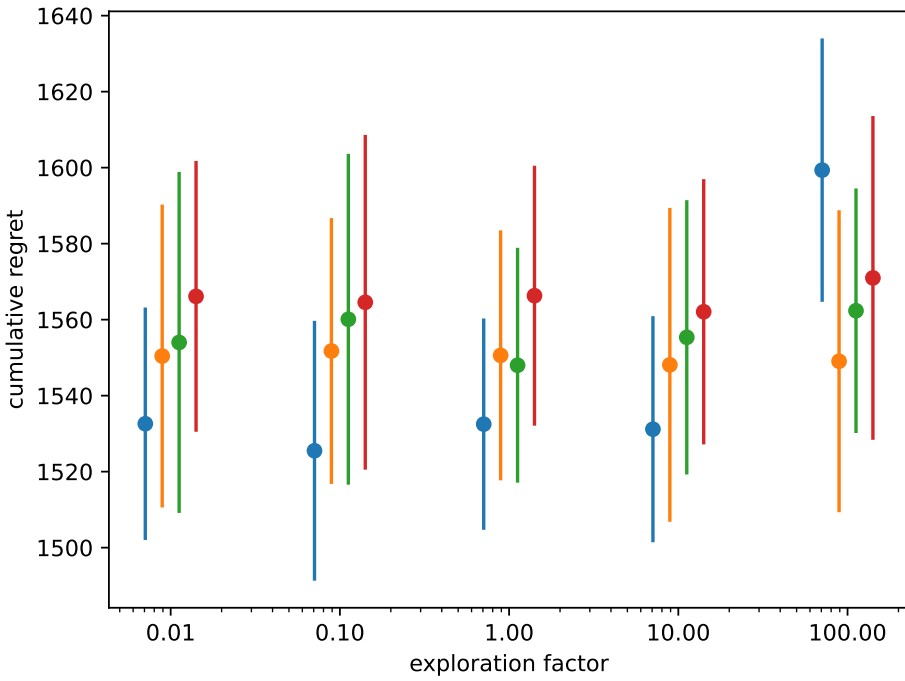

(b) Cumulative regret with varying exploration factors on the mushroom dataset

Figure 7: Comparison of TEUCB and TETS on the mushroom and the adult datasets with different exploration factors. Cumulative regret for a single experiment is calculated over 10,000 time steps and repeated 10 times. The results display the average cumulative regret plus/minus the standard deviation for each respective agent. Hyperparameters other than the exploration factor are selected as in Section 4.2. All agents are evaluated on the same exploration factors, but each agent is shifted slightly horizontally for visualization purposes.

