# OpenReview forum: "Tree Ensembles for Contextual Bandits"
_TMLR — Accepted by TMLR_

### Review · Reviewer_WzFc · 2024-08-26

**Summary Of Contributions:**

This work proposes a new framework for contextual multi-armed bandits based on several efficient tree ensemble methods, and it uses the classic upper confidence bound and Thompson sampling methods to draw the arm after tree fitting. Compared with some existing methods, the proposed algorithms can yield better performance in cumulative regret with less running time in a suite of benchmark datasets.

**Audience:**

Yes

**Broader Impact Concerns:**

No negative societal or ethical impact.

**Claims And Evidence:**

No

**Requested Changes:**

It would be better if the authors could add some theoretical bounds for the proposed algorithms.

**Strengths And Weaknesses:**

Strengths:
1. This work is well written and easy to follow.
2. Using tree ensembles is an interesting idea to improve the efficiency of bandit algorithms.

Weaknesses:
The regret bound is missing, even under the unrealistic assumptions made in page 4. I think nowadays for bandit research it is indispensable to deduce a valid regret bound under some milder assumptions (i.e. no assumption on the leaves directly). Otherwise, any state-of-the-art supervised learning algorithms can be directly implemented before the UCB or TS, and hence the novelty is limited. On the other hand, all the existing tree-based bandit algorithms are somehow inefficient in practice since their invention have to deal with the theoretical regret bound simultaneously, which is the most difficult part of research. Therefore, it is not fair to compare the algorithms in this paper with those baselines directly.

---

> ### Author Response · Authors · 2024-09-01
> **Generality and applicability of our work, instead of focus on theoretical analysis in a restricted setting**
>
> We thank the reviewer for the fast and informative review. We address the absence of a theoretical regret bound in the following manner.
>
> - In this work, we focus on the generality and applicability of our framework. Our goal is to solve real-world bandit problems using methods that achieve superior performance in both regret minimization and computational runtime. To this end, we avoid limiting assumptions that, while enabling theoretical analysis in some cases, may not be practical in real-world scenarios.
>
> - We respectfully disagree with the statement that any state-of-the-art supervised model can be selected for our purpose and automatically yield the best performance for bandit problems. The method must satisfy certain crucial properties, e.g., to concentrate efficiently around optimal choices but not so fast in order to prevent converging to suboptimal choices.
>
> - An important aspect of our study is to demonstrate the effectiveness and superior performance of tree ensemble methods compared to recent deep neural network models, which are often considered state-of-the-art for bandit problems. We show that tree ensemble methods achieve better and more stable results in both regret minimization and computational runtime (and thus greater applicability). It is notable that the most sophisticated (and practical) deep learning-based bandit methods, such as those by Zhu & Van Roy (2023), also lack solid theoretical analysis.
>
> - Finally, in terms of novelty, to the best of our knowledge, this is the first work to propose tree ensembles for contextual bandits, both in standard and combinatorial settings, in a generic and non-restrictive manner. Notably, the use of tree ensembles has not been explored for combinatorial bandits before, and we demonstrate their effectiveness in the important real-world application of navigation over road networks, modeling this task as a combinatorial contextual bandit problem.

---

> > ### Comment · Reviewer_WzFc · 2024-09-19
> > **Thanks for your responses**
> >
> > I'd like to thank the authors for their detailed responses to my questions and concerns. I still have concerns about lacking of theoretical support in this work, since for the bandit community the theoretical analysis is very important, and hence it would be nice if the authors could possibly have some theory support for their algorithms. I acknowledge that your algorithms can achieve better performance with less computational time, but existing algorithms you used for comparison have solid theoretical guarantee, which makes the comparison a little bit unfair.
> >
> > Other reviewers point out the usage of TS-based methods, which is a good suggestion. Since some work showcases that TS can yield better practical results compared with UCB. e.g. [An empirical evaluation of thompson sampling, Chapelle et al.]

---

> ### Author Response · Authors · 2024-09-19
> **Some clarifications**
>
> We thank the reviewer for the feedback on our response. We would like to address the two comments as the following.
>
> - For clarity in comparing theoretical methods, we would like to emphasize that some practical neural network-based approaches (e.g., Zhu & Van Roy, 2023) also lack theoretical regret analysis. Like our methods, they primarily focus on practical applicability. Additionally, in our experiments, we include a wide range of competitors, specifically 11 different methods, which makes our comparison study quite comprehensive. To our knowledge, such an extensive comparison across multiple methods is relatively unique in the investigation of contextual bandit methods.
>
> - Regarding the comparisons with Thompson sampling, we have already integrated and adapted this method within our framework. All of our experiments include this method, and we consistently observe promising results with this method integrated into our framework. For example, in Figures 1 and 2, Thompson sampling is integrated into several different models.

---

### Review · Reviewer_2NeU · 2024-09-06

**Summary Of Contributions:**

The authors propose applying tree ensemble methods to model the rewards in contextual multi-armed bandits (MAB). They introduce two algorithms—Tree Ensemble Upper Confidence Bound (TEUCB) and Tree Ensemble Thompson Sampling (TETS)—that adapt tree ensemble models, such as XGBoost and random forests, to this setting. the paper is purely experimental and lacks formal theoretical guarantees, such as regret bounds, which are common in bandit literature. While it is unsurprising that tree ensemble methods can outperform neural-based approaches in certain settings, neural-based methods [3,4] have the advantage of offering theoretical guarantees, which this work does not provide.

**Audience:**

Yes

**Broader Impact Concerns:**

N/A.

**Claims And Evidence:**

Yes

**Requested Changes:**

**Major concerns**
* See the weaknesses section above.

**Minor concerns**
* The authors should cite [1] when talking about LinUCB.
*  While the authors explore UCB and Thompson Sampling, they do not investigate the possibility of extending their work to include BayesUCB [2], which could add further depth and relevance.

**References**
* [1] Yasin Abbasi-Yadkori, David Pal, and Csaba Szepesvari. Improved algorithms for linear stochastic
bandits. In NeurIPS, 2011.
* [2] Emilie Kaufmann, Olivier Cappe, and Aurelien Garivier. On Bayesian Upper Confidence Bounds for Bandit Problems. In AISTATS, 2012.

**Strengths And Weaknesses:**

**Strengths**
* The use of tree ensemble models for contextual bandit problems is a creative and practical idea.
* The methods appear to be more computationally efficient than the neural-based approaches [3,4].

**Weaknesses**
* The paper is purely experimental and lacks formal theoretical guarantees, particularly regret bounds for the proposed methods. This is a significant limitation compared to other bandit algorithms, where theoretical guarantees are often a key feature, e.g., [1,3,4]. Without theoretical support, it's unclear how well the methods generalize beyond the datasets tested. This raises concerns about the robustness of the approach in different scenarios or real-world applications.
* The author's makes strong assumptions about the independence of trees in the ensemble and the Gaussian distribution of their predictions, which may not hold in practice. This assumption seems to made by convenience for the following UCB/TS. The authors acknowledge this but do not sufficiently explore the impact of these assumptions.
* The pseudo-code in Algorithm 1 is overly basic. It would be beneficial for the authors to provide detailed pseudo-code for both TEUCB and TETS, including all the relevant hyperparameters, to make the methods clearer and easier to replicate.
* The paper lacks publicly available code, which is critical for verifying and reproducing the results, especially in an empirical study.
* The experiments, while useful, are limited in scope. Testing the methods on a broader set of datasets, particularly those with more challenging characteristics (e.g., delayed rewards [3]), would help assess the robustness and generalizability of the approach.
* A sensitivity analysis of key hyperparameters (e.g., tree depth, number of trees, exploration factor) would provide readers with more practical insights into the robustness of the methods.

**References**
* [1] Yasin Abbasi-Yadkori, David Pal, and Csaba Szepesvari. Improved algorithms for linear stochastic
bandits. In NeurIPS, 2011.
* [2] Emilie Kaufmann, Olivier Cappe, and Aurelien Garivier. On Bayesian Upper Confidence Bounds for Bandit Problems. In AISTATS, 2012.
* [3] Weitong Zhang, Dongruo Zhou, Lihong Li, and Quanquan Gu. Neural thompson sampling. In ICLR, 2021
* [4] Dongruo Zhou, Lihong Li, and Quanquan Gu. Neural contextual bandits with UCB-based exploration. ICML, 2020.

---

> ### Author Response · Authors · 2024-09-19
> **To address the reviewer's comments from various aspects**
>
> First of all, we thank Reviewer 2NeU for the constructive feedback. We address their concerns as follows:
>
> 1. Concerning the lack of theoretical regret bounds, please see the response to Reviewer WzFc.
>
> 2. We appreciate the suggestion to elaborate on the independence assumptions we have made. We have designated Section A.2 of the Appendix for this purpose.
>
> 3. Based on the feedback given here and in requested change #4 of Reviewer wcpK, we have added separate code blocks for the calculations of the leaf values, including the relevant hyperparameters, for the application of TEUCB and TETS on both XGBoost and Random Forest. This can be found in Appendix A.3. The pseudo-code for TEUCB and TETS are very similar, and have identical hyperparameters. However, depending on the particular tree ensemble methods used the leaf values which are used for the ensemble's prediction and uncertainty are obtained in different ways, which we hope to clarify with the provided additional code blocks. Please let us know if there is anything else that is unclear with the algorithms.
>
> 4. Since this study is carried out together with a company, we are not able to release the code used for the road network navigation task. By rewriting the code for the other experiments, however, we hope and plan to be able to release the code for those parts upon publication.
>
> 5. The primary reason for choosing the datasets used in this paper is that they serve as benchmark datasets in the original papers introducing NeuralUCB, NeuralTS, and TreeBootstrap. They, therefore, provide benchmark results directly against the current state-of-the-art models we consider, and indirectly with other bandit algorithms through previous results in the literature. Further, we appreciate the suggestions here and have performed additional experiments with delayed feedback for the tree-based methods on two of the standard contextual bandit settings, which are included in Appendix A.5 of the revised paper.
>
> 6. We have run additional experiments where we examine the robustness to changes of the tree depth, number of trees, and exploration factor for TEUCB and TETS on two of the data sets. The respective results are included in Appendix A.5 of the revised version.
>
> Minor concerns:
>
> 7. We agree that the referenced paper should be cited when discussing LinUCB, and have updated the paper accordingly.
>
> 8. We appreciate the suggestion of BayesUCB and agree that it would be interesting to evaluate, but will have to leave this item for future work. BayesUCB is less common than UCB, and on the other hand, in this paper, we focus on the non-Bayesian setting.

---

### Review · Reviewer_wcpK · 2024-09-08

**Summary Of Contributions:**

This paper presents novel methods for constructing confidence intervals around the predictions of tree-based classifiers and regression models, and apply these to contextual bandit problems, as well as a "structured" combinatorial bandit problem.  They present two methods for constructing (Gaussian) posteriors over the outputs of tree models, one for XGBoost trees, and one for Random Forests (RF).  They then apply these posteriors for both upper confidence bound (UCB) and Thompson sampling (TS) exploration in bandit settings.  They compare these methods against previous contextual bandit methods that use linear, neural, and tree-based (TreeBootstrap) classifiers, and find that they generally outperform these baselines in terms of total accumulated utility (or cost).

**Audience:**

Yes

**Broader Impact Concerns:**

I identified no future ethical concerns arising from this work that would need to be addressed in the text.

**Claims And Evidence:**

No

**Requested Changes:**

1. In Section 4, for each experiment, indicate how many initial samples $T$ were used to initialize the models
2. Also, for the experiments in Section 4.2, estimate the *exploration rate* as a function of time.  This is the fraction of contexts in which the learner chooses an arm that does not maximize the expecteed return predicted by the model.  Also, estimate the error rate of the learner model as a function of time.  It is currently difficult to assess whether differences in performance are due to the amount of time it takes for exploration to stop altogether, or whether it has more to do with the accuracy of the learned models themselves.
3. Check that the travel time predictions in Section 4.3 are lower-bounded by some reasonable value (or even just clipped to be non-negative).  Also, explain whether figure 3 shows a single sampled path for each learner, or the total set of road segments explored by those methods throughout the learning process.
4. Include separate algorithm blocks for the construction of the posterior distributions for XGBoost and RF, as it appears that the pseudocode in algorithm 1 may not cover XGBoost exactly as it is described in the text.  Also, describe these algorithms separately in text.
5. Include short explanations of XGBoost and Random Forests in the Background section (Section 2).

**Strengths And Weaknesses:**

**Strengths:**

1. The methods for estimating posterior distributions around tree classifiers are novel and interesting, even if they are only heuristics.
2. Their experiments are thorough, both in terms of tasks and baseline methods against which they compare, but see "weaknesses" for the limitations of these experiments

**Weaknesses:**

1. The biggest weakness is in section 4.3, which presents the results of TEUCB and TETS in the Luxembourg roads domain.  While the proposed methods appear to perform substantially better in this setting, I have a number of concerns about the validity of the results.  Looking at figure 3 in appendix A.2, it appears that none of the methods are sampling sensible routes between the given start and end locations.  In addition to being far longer than the optimal path shown in figure 3a, they appear to contain loops, which should be impossible if they were generated using Dijkstra's algorithm.  I can think of two possible explanations for this:
    1. These figures (aside from 3a) show *all* road segments sampled by each of the learners during the training process (if so, this needs to be made clear)
    2. There is no restriction on the models preventing them from predicting *negative* edge lengths, causing their implementation of Dijkstra's algorithm to return nonsensical looping paths
2. The contextual bandit results in section 4.2 are also problematic.  In all but one of the tasks (the shuttle task), the number of classes, and therefore the number of arms, is only 2, suggesting that the exploration problem is somewhat limited.  Interestingly, the one problem where the proposed TEUCB and TETS methods are outperformed by the existing TreeBootstrap method is also the one problem in Section .42 with more than 2 arms, where improved exploration should be most helpful.  See the "requested changes" section for suggestions on how these issues could be addressed.
3. The methods for building posterior distributions around the XGBoost and Random Forest classifiers are *heuristic*, that is, they would not accurate under any reasonable assumption about the data generating process.  While this is not an issue itself, the heuristic nature of the estimates should be stated explicitly, and the methods should be presented separately for the XGBoost and RF classifiers, as these are very different methods.

---

> ### Author Response · Authors · 2024-09-19
> **To address the reviewer's comments from various aspects**
>
> We thank reviewer wcpK for the thoughtful comments. We address the requested changes in the following way.
>
> 1. The number of initial samples is set to 10 x the number of arms for all of the experiments on UCI datasets, and to 10 paths in the navigation task. In the revised version, we have included this information.
>
> 2. Regarding the experiments in section 4.2, which you also mention in Weaknesses (bullet point 2), the primary reason for choosing the datasets used in this paper is that they serve as benchmark datasets in the original papers introducing NeuralUCB, NeuralTS, and TreeBootstrap. They, therefore, provide benchmark results directly against the current state-of-the-art models we consider, and indirectly with other bandit algorithms through previous results in the literature. Also, although TreeBootstrap performed slightly better than TEUCB and TETS on the Shuttle data set, the difference is not significant. Furthermore, the most challenging data set in our experiments is the real-world road network which consists of a much larger set of arms. Although the setup is different, as it is a combinatorial contextual problem with semi-bandit feedback, we argue that this experiment provides evidence of the advantage of TEUCB and TETS over TreeBootstrap for hard bandit problems with many arms. Finally, please note that we have added a number of novel experimental studies in Appendix A.5, investigating various aspects of the proposed methods.
>
> 3. As the edge weights in this problem setting represent travel time, they should indeed be positive. This has been checked and confirmed. We understand your concerns here and thank you for pointing out that it had not been made clear that Figure 3 shows the total set of road segments explored by the respective algorithms throughout the learning process. We clarify this in the revised version.
>
> 4. We agree that it is a nice addition to the paper to include separate code blocks describing the procedure for XGBoost and Random Forest individually. We have included them in Appendix A.3 of the revised version.
>
> 5. We have introduced and described XGBoost and Random Forest better in Sections 3.6 and 3.7 respectively. It was a deliberate choice not to put an emphasis on XGBoost and Random Forest in the Background section which could mislead readers into believing TEUCB and TETS are specific for these methods, whereas in reality they can be used with any type of tree ensemble.

---

### Decision · Action_Editor_Cs6G · 2024-09-29

**Recommendation:** Accept with minor revision

**Comment:**

The only main concern that some reviewers continued to have after the discussion phase, is the lack of theoretical guarantees (e.g., bounds on regret), which are often present in work on multi-armed bandit algorithms. However, my reading of the acceptance criteria of TMLR imply that this should not block acceptance: the paper as is has interesting and relevant contributions, describes appropriate experiments to evaluate them, and these experiments provide adequate support for the claims that are made.

There are some concerns around the extent to which the results would generalise to different datasets/problems (as there are no theoretical guarantees we can rely on), but I see quite consistent results across 5 different datasets/problems, which I view as reasonable support.

---

**I do want to request the following revisions from the authors**:
- Judging from the last paragraph of 4.3.2, on the more complex real-world task, the newly proposed algorithms use a similar amount of computation time to the baselines, not less. In light of this, the claim "our methods benefit from more effective learning with less computational overhead" could use more nuance.
- Given the prevalence of theoretical analyses in work on Multi-Armed Bandits, it would be appropriate to signal to potential readers that your evaluation of the algorithms is solely empirical. Doing this in the Introduction could be appropriate.
- The conclusion is very short. It summarises what types of experiments were run, but does not even summarise what the outcomes were of those experiments and what conclusions follow from that. This should be added. It could also be useful to again acknowledge the limitation of the lack of theoretical results, and suggest this as a potential direction for future research.
- Try to do another round of proofreading for typos. I spotted at least one: "Asspendix" at the bottom of page 4.

Note that, for this revision, it will not be necessary to stick to 12 pages. It is fine to run a bit over this number of pages.

**Audience:**

All reviewers agree that at least some of the TMLR audience would be interested in this work.

**Claims And Evidence:**

Experiments across four standard datasets, plus one more complex real-world task, as well as experiments used to analyse the sensitivity to hyperparameter values, provide adequate support for the claims made in the paper.